# Quantitative proteomics reveals key roles for post-transcriptional gene regulation in the molecular pathology of facioscapulohumeral muscular dystrophy

Sujatha Jagannathan[1,2,3†]*, Yuko Ogata[4], Philip R Gafken[4], Stephen J Tapscott[3]*, Robert K Bradley[1]*

[1]Computational Biology Program, Public Health Sciences Division, Fred Hutchinson Cancer Research Center, Seattle, United States; [2]Basic Sciences Division, Fred Hutchinson Cancer Research Center, Seattle, United States; [3]Human Biology Division, Fred Hutchinson Cancer Research Center, Seattle, United States; [4]Proteomics Shared Resource, Fred Hutchinson Cancer Research Center, Seattle, United States

*For correspondence:
sujatha.jagannathan@ucdenver.
edu (SJ);
stapscot@fredhutch.org (SJT);
rbradley@fredhutch.org (RKB)

Present address: †Department of Biochemistry and Molecular Genetics & RNA Bioscience Initiative, University of Colorado Denver School of Medicine, Aurora, United States

Competing interests: The authors declare that no competing interests exist.

**Abstract** DUX4 is a transcription factor whose misexpression in skeletal muscle causes facioscapulohumeral muscular dystrophy (FSHD). DUX4's transcriptional activity has been extensively characterized, but the DUX4-induced proteome remains undescribed. Here, we report concurrent measurement of RNA and protein levels in DUX4-expressing cells via RNA-seq and quantitative mass spectrometry. DUX4 transcriptional targets were robustly translated, confirming the likely clinical relevance of proposed FSHD biomarkers. However, a multitude of mRNAs and proteins exhibited discordant expression changes upon DUX4 expression. Our dataset revealed unexpected proteomic, but not transcriptomic, dysregulation of diverse molecular pathways, including Golgi apparatus fragmentation, as well as extensive post-transcriptional buffering of stress-response genes. Key components of RNA degradation machineries, including UPF1, UPF3B, and XRN1, exhibited suppressed protein, but not mRNA, levels, explaining the build-up of aberrant RNAs that characterizes DUX4-expressing cells. Our results provide a resource for the FSHD community and illustrate the importance of post-transcriptional processes in DUX4-induced pathology.

## Introduction

Facioscapulohumeral muscular dystrophy (FSHD) is caused by the inappropriate expression of an early embryonic transcriptional activator, DUX4, in adult muscle, leading to cell death (*Tawil et al., 2014*; *Lemmers et al., 2010*). Decades of work have generated a detailed list of the genes and pathways affected by DUX4 that may underlie FSHD pathophysiology (*Geng et al., 2012*; *Block et al., 2013*; *Young et al., 2013*; *Banerji et al., 2015*; *Feng et al., 2015*; *Homma et al., 2015*; *Dmitriev et al., 2016*; *Shadle et al., 2017*). An integrated model for how those DUX4-induced changes lead to disease has, however, remained elusive (*Campbell et al., 2018*; *Lek et al., 2015*; *Tassin et al., 2013*). As transient and pulsatile expression of DUX4 is sufficient to induce pathology and cell death (*Rickard et al., 2015*), it is critical that we understand the cellular events and pathways set in motion by DUX4 that lead to eventual cell death, in order to develop effective therapeutics for FSHD.

DUX4 induces changes in the expression of hundreds of genes that impact dozens of highly interconnected pathways (*Geng et al., 2012*; *Block et al., 2013*; *Young et al., 2013*; *Banerji et al.,*

*2015*; *Feng et al., 2015*; *Homma et al., 2015*; *Dmitriev et al., 2016*; *Shadle et al., 2017*), making a cause-and-effect relationship between dysregulated gene expression and FSHD pathology difficult to discern. Because DUX4 is a strong transcriptional activator, most studies of DUX4 activity have focused on measuring gene expression at the transcript level (*Geng et al., 2012*; *Rickard et al., 2015*; *Knopp et al., 2016*), thereby implicitly assuming that the transcriptome accurately represents the cellular proteome in DUX4-expressing cells. Although this is a reasonable assumption, it is well known that RNA and protein levels are not always concordant and that post-transcriptional regulation can result in divergent RNA and protein levels (*Schwanhäusser et al., 2011*). A few proteomics studies have been conducted on FSHD muscle biopsies, but these early studies lack the depth necessary to allow meaningful comparisons with the DUX4-induced transcriptome (*Tassin et al., 2012*; *Celegato et al., 2006*; *Laoudj-Chenivesse et al., 2005*). Furthermore, given our recent discovery that DUX4 induces the proteolysis of a key RNA-binding protein, UPF1 (7), we hypothesized that paired measurements of RNA and protein levels might be particularly important in identifying altered post-transcriptional gene regulation in DUX4-expressing cells. Hence, we set out to generate reliable RNA- and protein-level measurements of DUX4-induced gene expression and, thereby, to elucidate the extent of post-transcriptional gene dysregulation in DUX4-expressing cells.

We used our previously established and validated cell culture models of DUX4 expression (*Jagannathan et al., 2016*) to conduct RNA-seq and Stable Isotope Labeling with Amino acids in Cell culture (SILAC) coupled with quantitative mass spectrometry (*Harsha et al., 2008*). The resulting data enabled us to measure DUX4-induced alterations in the transcriptome as well as protein levels for ~4000 genes with high confidence. Comparison of the transcript-level and protein-level alterations revealed three distinct patterns of expression for different subsets of genes: 1) concordant changes in expression at the RNA and protein levels for many transcriptional targets of DUX4; 2) post-transcriptional buffering of the expression of many genes, especially of those involved in stress-response pathways; and 3) discordant gene expression changes at the RNA versus protein levels for many genes, particularly those involved in RNA surveillance. Together, these findings highlight the importance of measuring the expressed proteome in order to understand DUX4 biology and the FSHD disease process fully.

## Results

### Determining protein-level alterations in DUX4-expressing cells through quantitative mass spectrometry

In order to measure DUX4-induced changes to the cellular proteome, we conducted SILAC-based mass spectrometry in two independent DUX4 expression systems (*Figure 1A*). We previously showed that comparable gene expression profiles, which accurately capture the transcriptome of FSHD cells, are produced when DUX4 is expressed either via a lentiviral vector or when an inducible transgene is integrated into the genome of a myoblast cell line (*Jagannathan et al., 2016*). Here, we used both of these expression systems to corroborate our results internally, thereby ensuring that our proteomic data were robust with respect to choice of model system.

We first established the efficiency of SILAC labeling and determined appropriate cell culture conditions and experimental timing in order to ensure near-complete labeling of proteins with heavy isotopic arginine and lysine. To this end, we cultured immortalized MB135 myoblasts in SILAC labeling media for a week and subjected the total protein from labeled cells, as well as a 1:1 mix of labeled to unlabeled cells, to mass spectrometry. Using the resulting mass spectra, we calculated labeling efficiency in two ways. First, we quantified the relative abundance of heavy and light spectra for several individual peptides and found that it showed a > 95% labeling efficiency (a representative example is shown in *Figure 1—figure supplement 1A*). Next, we analyzed the proteome-wide distribution of log-transformed heavy to light ratios of peptides from both the heavy-only (*Figure 1—figure supplement 1B*) and the heavy:light mixed samples (*Figure 1—figure supplement 1C*). The distribution of heavy-to-light ratios was strongly skewed to the right in the heavy-only samples, indicating robust labeling with the heavy amino acids. By contrast, the heavy:light mixed sample yielded a distribution centered roughly around zero, as expected. On the basis of these data, we proceeded with the labeling conditions and increased the labeling time even further (to three weeks) in order to achieve maximal SILAC labeling of the MB135 myoblasts.

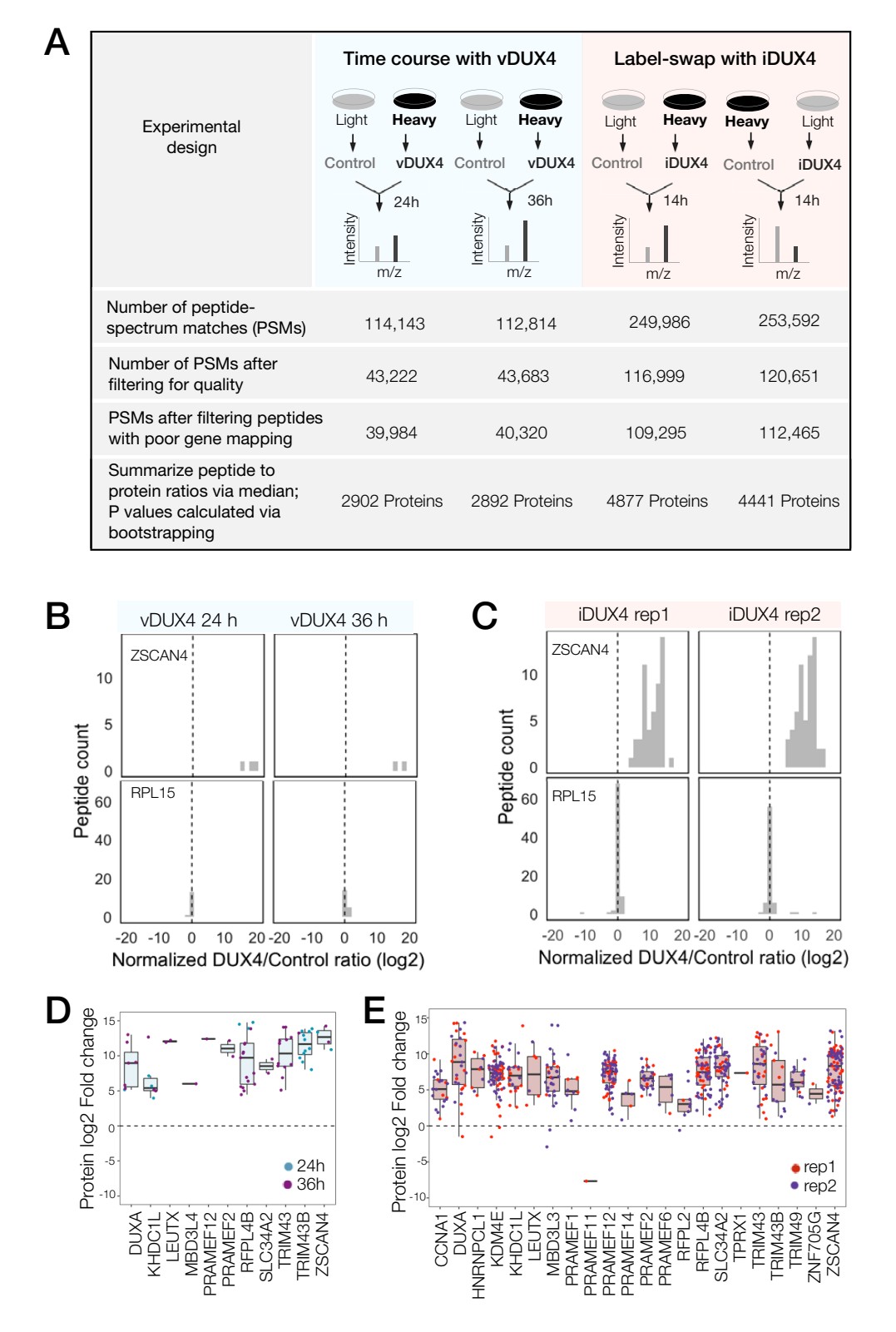

**Figure 1.** Quantitative mass spectrometry of DUX4-expressing cells. (A) Schematic of the experimental set up and the subsequent data analysis steps used to measure protein fold change in cells expressing vDUX4 or iDUX4. (B) Histogram of normalized, log$_2$-transformed DUX4/Control ratios for all peptides mapping to ZSCAN4, a DUX4 target gene (top panels), and for RPL15, a housekeeping gene (bottom panels), following 24 and 36 hr of vDUX4 expression. (C) Histogram of normalized, log$_2$-transformed DUX4/Control ratios for all peptides mapping to ZSCAN4, a DUX4 target gene (top

*Figure 1 continued on next page*

*Figure 1 continued*

panels), and for RPL15, a housekeeping gene (bottom panels), in the two label-swap replicates of iDUX4 expression. (D) Box plot of normalized, log$_2$-transformed DUX4/Control ratio for peptides corresponding to the FSHD biomarker transcripts identified by *Yao et al. (2014)* in the vDUX4 dataset. Each dot refers to an individual peptide that was quantified and the color represents the time point of vDUX4 expression (24 hours in blue and 36 hours in purple). (E) Box plot of normalized, log$_2$-transformed DUX4/Control ratio for peptides corresponding to the FSHD biomarker transcripts identified by *Yao et al. (2014)* in the vDUX4 dataset. Each dot refers to an individual peptide that was quantified and the color denotes replicate 1 (red) or replicate 2 (purple) of iDUX4 expression.

The online version of this article includes the following figure supplement(s) for figure 1:

**Figure supplement 1.** Demonstration of peptide to protein summarization for two candidate genes.

MB135 myoblasts that had been adapted to light or heavy SILAC media for 3 weeks were transduced with lentivirus carrying DUX4 (vDUX4) or GFP (vGFP) expression constructs. Samples were collected 24 hr and 36 hr post-transduction. In an independent experiment, MB135 cells carrying a doxycycline-inducible DUX4 transgene (iDUX4; *Jagannathan et al., 2016*) were adapted to SILAC media for 3 weeks before DUX4 expression was induced with 1 μg/ml of doxycycline for 14 hr in two replicates carrying heavy and light SILAC labels. Paired controls with no treatment were also collected with both heavy and light labels. Total protein from cells expressing DUX4 were mixed with an equal amount of total protein from cells without DUX4 expression containing the opposite SILAC label to generate samples that were then subjected to mass spectrometry.

Peptide-spectrum matches (PSMs) with quantified heavy to light ratios were subject to thorough screening for quality (e.g., filtering out single-peak spectra and spectra without unique mapping; *Figure 1A*; see Materials and methods for further details). A histogram of log-transformed DUX4 to Control abundance ratios (log$_2$ (DUX4/Control) ratio) of peptides mapping to a DUX4 target gene, ZSCAN4, from both vDUX4 and iDUX4 datasets showed highly skewed log$_2$ (DUX4/Control) ratio, consistent with significant upregulation of the protein upon DUX4 expression (*Figure 1B–C*). By contrast, plotting the log$_2$ (DUX4/Control) ratio of all individual peptides mapping to the housekeeping gene RPL15 showed that the ratio is centered around zero (*Figure 1B–C*), as would be expected for a gene with no differential expression upon DUX4 induction. These example plots illustrate the strong agreement between the expected and observed protein fold change values determined by SILAC mass spectrometry. Moreover, of the 65 genes identified by *Yao et al. (2014)* as potential FSHD biomarkers on the basis of transcriptome analysis of FSHD patient samples, 8 were quantified in the vDUX4 proteomics study and 25 were quantified in the iDUX4 proteomics study and all of these genes show high induction at the protein level (*Figure 1D,E*). Note the lower number of peptides (and hence quantified proteins) from the vDUX4 sample, which indicates the lower depth of this dataset and yet yields fold changes that are highly consistent with those from the higher-depth iDUX4 dataset.

## Assessing the concordance of fold change in RNA- and protein-expression

Next, using the iDUX4 dataset, we performed peptide to protein summarization by measuring the median heavy/light ratios of all of the peptides mapping to a certain protein in both label-swap replicates to obtain gene-level log$_2$ (DUX4/Control) ratios (*Figure 1A* and *Figure 1—figure supplement 1D–E*; *Supplementary file 1*). After filtering out genes that were only observed in one of the two label-swap replicates, we obtained quantitative proteomics information for 4005 genes, 3961 of which also had a corresponding RNA-seq measurement (*Figure 2A*; RNA-seq data previously reported by *Jagannathan et al., 2016*). The lower number of genes quantified by proteomics compared to RNA-seq is expected as proteomics is known to have lower sensitivity than RNA-seq.

To compare the RNA and protein expression level changes upon DUX4 expression qualitatively, we assessed the overlap of genes with an expression change of 4-fold or above. Among genes that are upregulated (>2 log$_2$ fold change), the concordance between RNA and protein was roughly 40–50%, whereas similarly downregulated genes show very little concordance (*Figure 2A*). To obtain a more quantitative measure of concordance, we generated a scatter plot of the RNA versus protein fold change for the 3961 genes (*Figure 2B*). We found a reasonable level of correlation between these values with a Pearson's correlation coefficient, r, of 0.51 (p-value<2.2e-16).

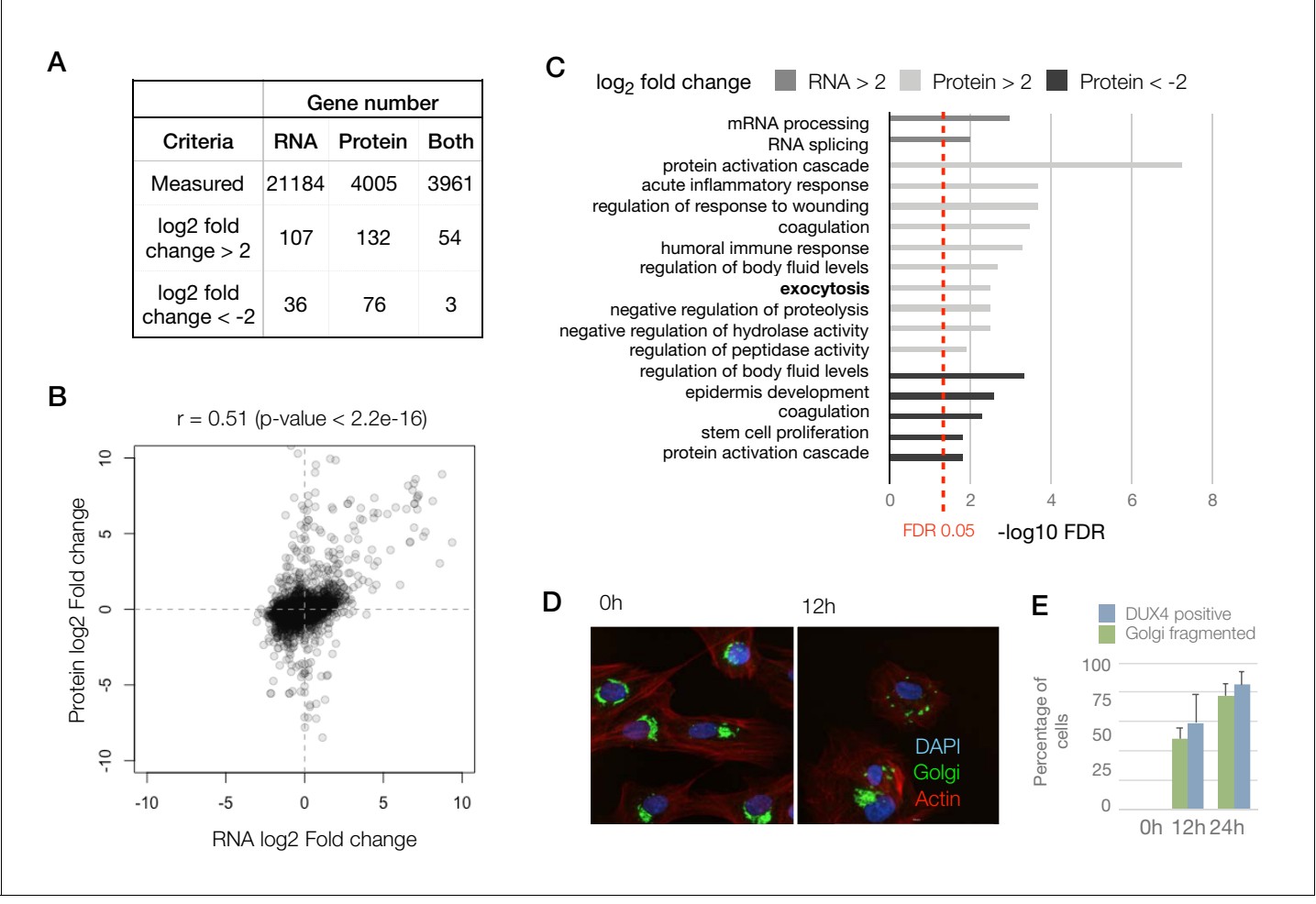

**Figure 2.** Concordant changes in the RNA and protein abundance of several DUX4 transcriptional targets. (A) Table showing number of genes with fold changes in their RNA and protein levels in the iDUX4 datasets. (B) Scatter plot of log$_2$ fold change in RNA levels (DUX4/control) versus log$_2$ fold change in protein levels (DUX4/control) for all genes. (C) Gene Ontology analysis of genes that are up- or downregulated > 4-fold at the RNA or protein level. Multiple testing correction was performed using a Benjamini–Hochberg procedure. (D) Fluorescence micrographs of cells with and without DUX4 induction for 14 hr, stained with DAPI (blue), anti-GM130 (Golgi apparatus; in green) and phalloidin-555 (actin; in red). (E) Quantification of the percentage of cells with DUX4-positive nuclei and fragmented Golgi apparatus at 12 hr and 24 hr post doxycycline induction of DUX4 expression. Error bars represent standard deviations.

To assess if other pathways are affected similarly at the RNA versus protein levels, we conducted a Gene Ontology (GO) analysis searching for genes that are up- or downregulated at the RNA and protein levels (*Figure 2C*). Surprisingly, we observed that the pathways that are affected at the RNA versus protein levels are quite distinct. Transcript level changes occur in genes that are involved in transcription and mRNA processing, whereas protein-level changes impact pathways including the humoral immune response, proteolysis and exocytosis. The exocytosis pathway was not implicated in any of the previous studies of DUX4 gene expression, so we sought to examine this phenomenon further by imaging the Golgi apparatus, which is the source of exocytotic vesicles in the cell (*Rodriguez-Boulan and Müsch, 2005*). We found that DUX4-expressing cells showed severe fragmentation of their Golgi apparatus, which could be an indicator of a perturbation in the cellular secretory pathways (*Bexiga and Simpson, 2013*) (*Figure 2D*). As not every cell in the DUX4-induced condition showed fragmented Golgi apparatus, we also quantified the percentage of cells with robust DUX4 expression at 12 hr and 24 hr post-induction using immunostaining. We found the percentages of cells with DUX4 expression to be comparable to the percentage of cells with fragmented Golgi (*Figure 2E*).

Taken together, these results demonstrate that analyzing the protein measurements may give us insights that were not discernable in the transcriptome fold-change analysis performed in earlier studies.

## Post-transcriptional buffering of stress-response genes may exacerbate DUX4 toxicity

Although many of the genes that are induced at the transcript level are largely also induced at the protein-level, a subset of genes showed no change in their protein level while their transcripts were up- or downregulated to a significant degree (678 genes, shaded blue in *Figure 3A*), indicating post-transcriptional buffering of the protein levels. Most notably, several housekeeping genes that respond to protein folding stress or dsRNA-induced stress showed transcriptional upregulation with minimal protein-level upregulation (*Figure 3B*).

Given that both unfolded protein and dsRNA-induced stresses converge in the phosphorylation of eIF2a and lead to translation inhibition (*Cláudio et al., 2013*), we asked whether the timing of the transcription of various stress-response genes coincideswith their translational downregulation. We found that HSPA5, a prominent marker of the unfolded protein response pathway (*Oslowski and Urano, 2011*), shows transcriptional upregulation during a time period that temporally coincides with eIF2a phosphorylation and the reduced incorporation of [35 S]-labeled methionine, a proxy for bulk translation efficiency (*Figure 3C–E*). These data demonstrate that translation inhibition caused by various cellular stresses and the resulting post-transcriptional buffering prevents DUX4-expressing cells from mounting a robust stress response.

## Post-transcriptional modulation of RNA quality control pathway by DUX4

Next, we focused our analysis on the subset of genes that showed significant changes at the protein level with either no change or a change in the opposite direction in their transcript abundance (198 genes shown as 'gold' circles in *Figure 4A*). Pathway analysis did not reveal any significant trends among these genes. So instead, we decided to focus on one of the pathways that we have previously shown to be post-transcriptionally modulated, the nonsense-mediated RNA decay (NMD) pathway (*Feng et al., 2015*).

A diagram showing RNA- versus protein-level changes in various components of this pathway demonstrates substantial post-transcriptional regulation in this pathway (*Figure 4B*). Many of these genes, including UPF1, UPF2, UPF3B and XRN1, showed downregulation at the protein level. The downregulation of XRN1 is of particular interest as it is the 5′−3′ exonuclease that degrades NMD targets upon cleavage by the endonuclease, SMG6 (*Palacios, 2013*). Moreover, SMG6 too is downregulated to a $\log_2$ fold change of −4.7, although it is only detected as a single peptide and hence was filtered out of our analysis. Thus, DUX4-induced NMD inhibition appears to be a result of the post-transcriptional downregulation of multiple key players of the NMD pathway, which explains the severity of NMD inhibition in DUX4-expressing cells.

Post-transcriptional downregulation of a gene can be achieved via two means: reduced translation or increased protein degradation. We have previously shown that DUX4 induces proteasome-mediated degradation of UPF1 (*Feng et al., 2015*). Hence, we asked whether DUX4 affects known components and regulators of the ubiquitin proteasome. A scatterplot of all ubiquitin proteasome regulators shows a change in the expression of several such genes, one or more of which may underlie the rapid degradation of UPF1 (*Figure 4C*). Further studies are needed to reveal the precise molecular mechanism behind this regulatory pathway and its downstream consequences. In summary, we propose that post-transcriptional gene regulation plays a critical role in inhibiting NMD and in perturbing the proteostasis in DUX4-expressing cells, and thus may underlie key aspects of FSHD pathology (*Figure 4D*).

Finally, in order to enable researchers and patients in the FSHD community to access the data generated in this study, we developed a web tool for easy visualization of these data (screenshot shown in *Figure 5*). This tool can be freely accessed at https://dynamicrna.shinyapps.io/dataviz/.

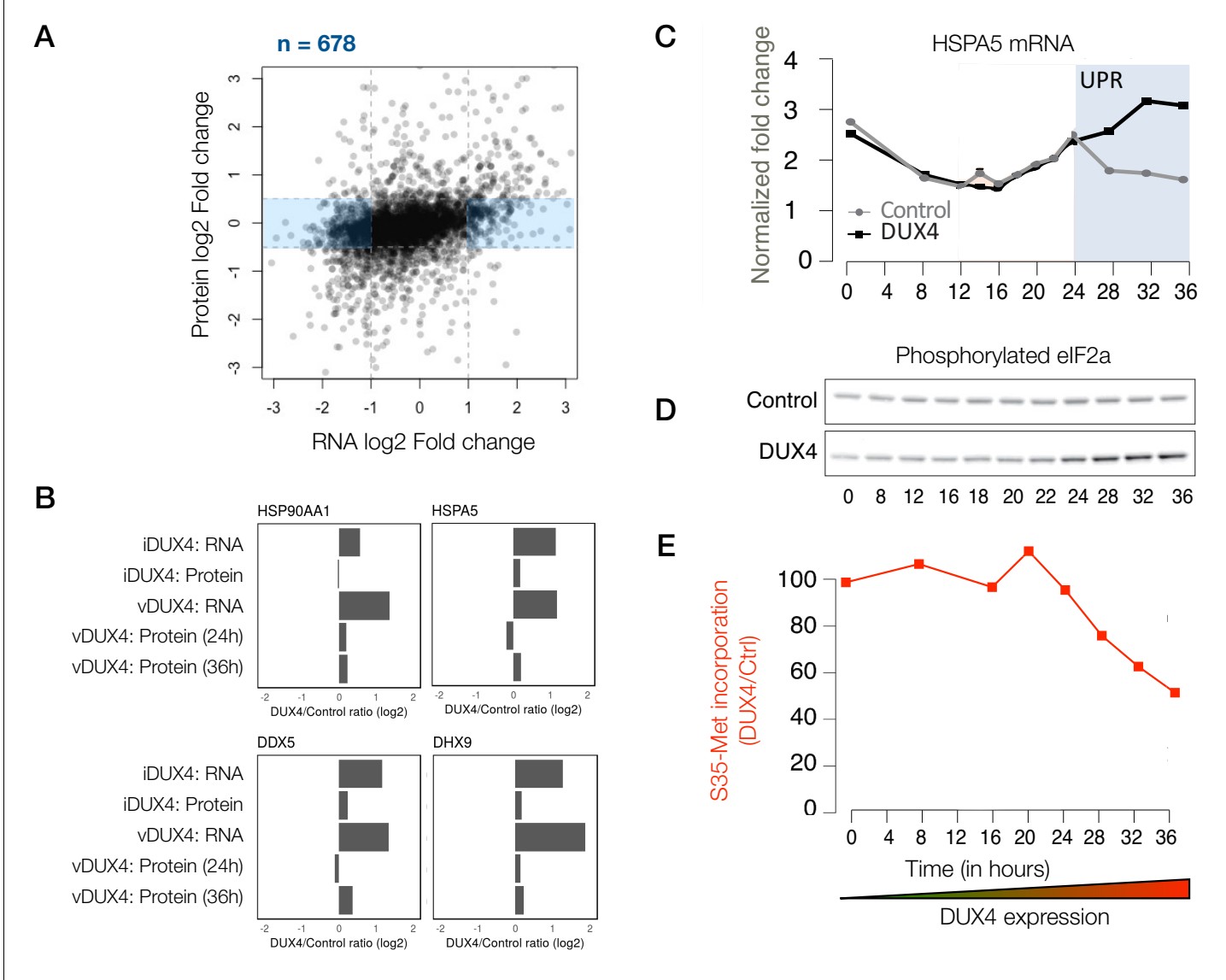

**Figure 3.** Extensive post-transcriptional buffering of stress-response genes in DUX4-expressing cells. (A) Scatter plot of $\log_2$ fold change in RNA levels (DUX4/control) versus $\log_2$ fold change in protein levels (DUX4/control). Genes that are upregulated at the RNA level but no significant change at the protein level fall within the blue rectangle. (B) RNA and protein fold change across the different datasets for representative genes that show post-transcriptional buffering. (C) Changes in the RNA levels of the chaperone HSPA5 over the duration of vDUX4 expression (or control expression) as measured by qRT-PCR. The region shaded in blue represents the region where the vDUX4 sample significantly deviates from the Control because of the induction of the unfolded protein response (UPR). (D) Levels of phosphorylated eIF2$\alpha$ in control cells versus DUX4-expressing cells over an expression time course as detected by immunoblots. (E) Percentage of [$^{35}$S]-Methionine incorporation (representing translation efficiency) in DUX4-expressing cells normalized over control cells during the time course of DUX4 expression.

## Discussion

Most of the highly induced DUX4 transcriptional targets are germline and early embryonic genes that are normally never expressed in adult muscle (*Geng et al., 2012*). So, it is possible that despite being expressed at the transcript level, such genes may be translated poorly and/or be degraded rapidly upon translation because of the lack of cell-type chaperones or other factors. Here, we used quantitative mass spectrometry in two different cell culture models of FSHD to demonstrate that DUX4-induced transcripts are efficiently translated into stable proteins in muscle cells. We demonstrated previously that our vDUX4 and iDUX4 systems accurately capture the transcriptional program of FSHD cells (*Jagannathan et al., 2016*), so it is reasonable to assume that DUX4-induced

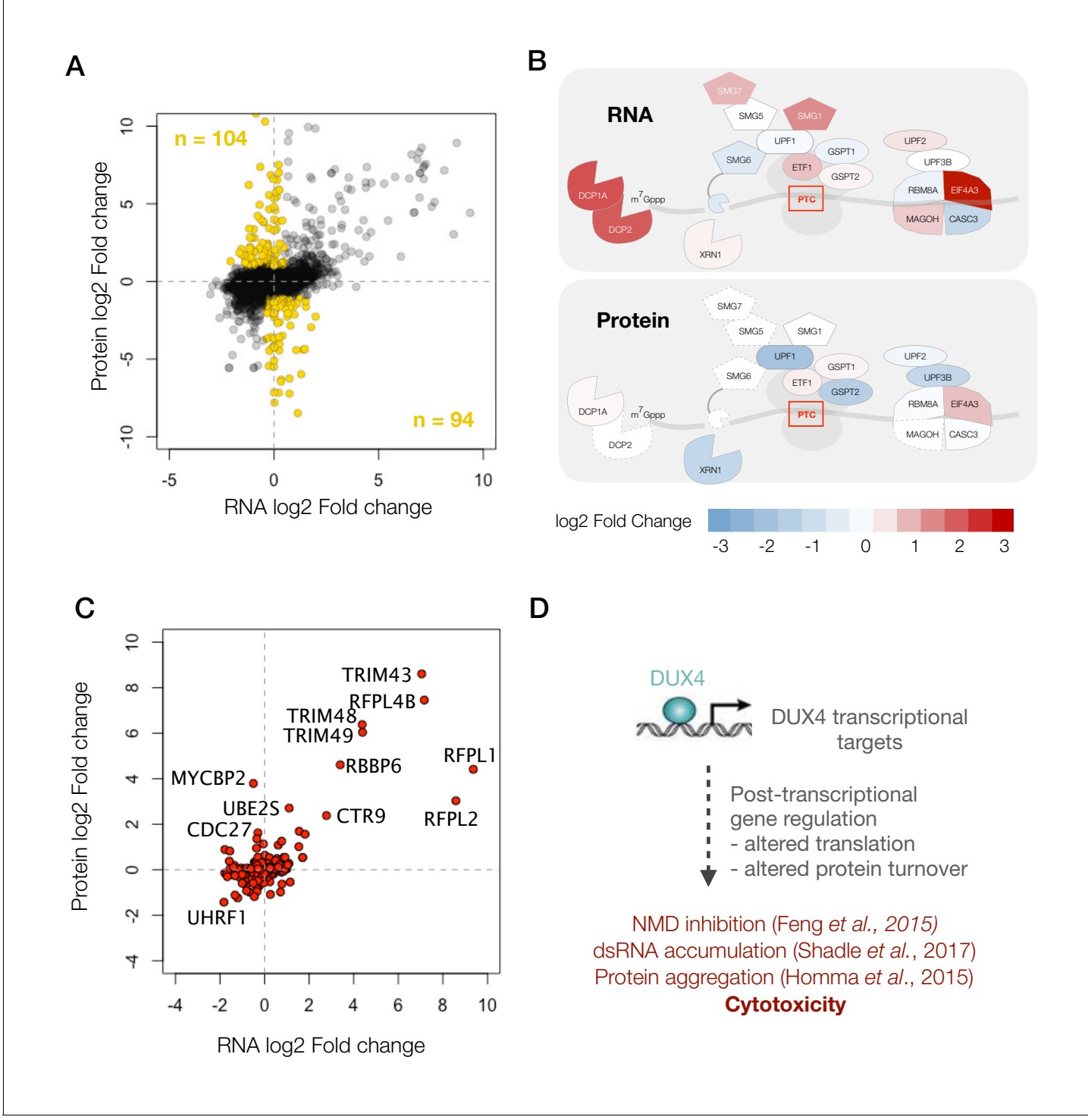

**Figure 4.** DUX4 induces post-transcriptional gene regulation. (**A**) Scatter plot of $\log_2$ fold change in RNA levels (DUX4/control) versus $\log_2$ fold change in protein levels (DUX4/control). The post-transcriptional regulation of genes results in drastic changes in protein level without major changes in RNA levels (highlighted in gold). (**B**) Schematic representation of RNA- and protein-level changes for the genes involved in mRNA surveillance. Colors represent a heat map of actual fold changes in RNA levels (top) and in protein levels (bottom). Proteins with fewer than two quantified peptides are outlined by dotted lines. (**C**) Scatter plot of $\log_2$ fold change in RNA levels (DUX4/control) versus $\log_2$ fold change in protein levels (DUX4/control) for genes in the ubiquitin proteasome pathway. (**D**)Model for DUX4-induced post-transcriptional gene regulation.

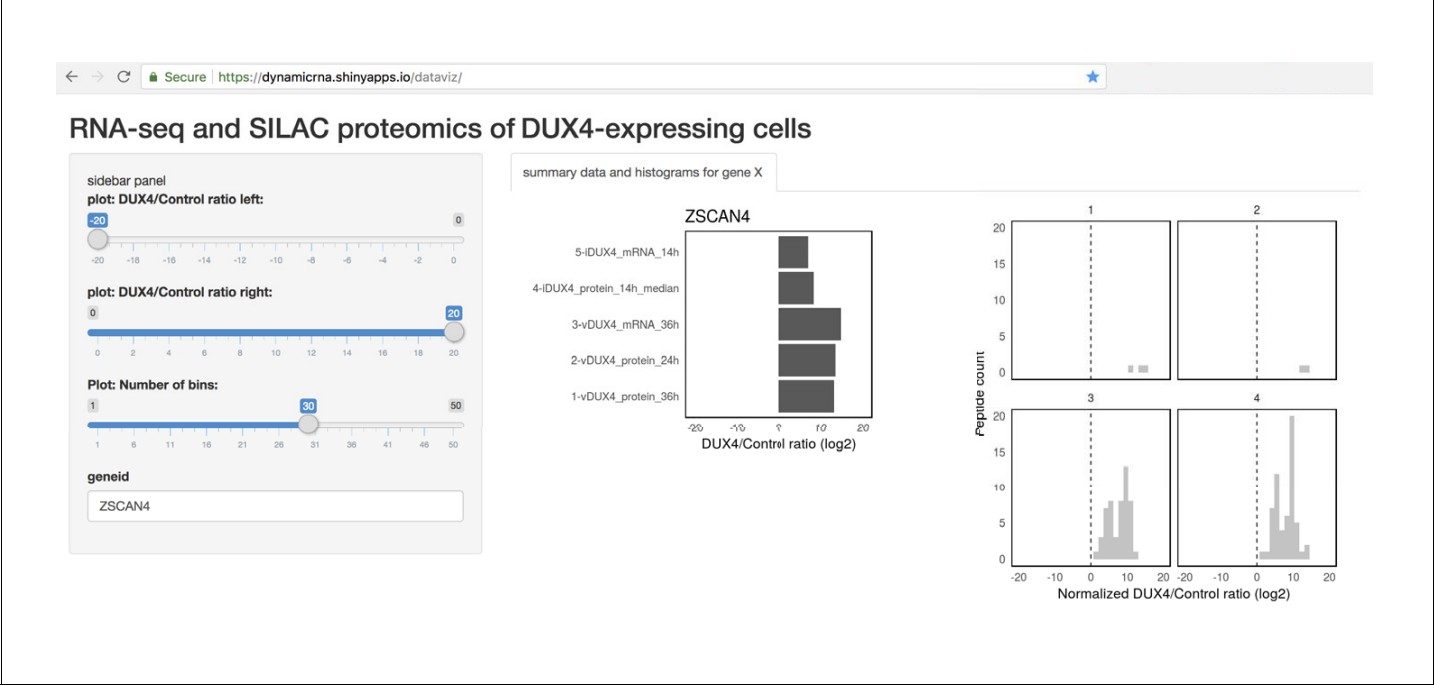

**Figure 5.** Tool allowing easy access to the data generated in this study. Screenshot of the Shiny web server showing a sample analysis of the RNA- and protein- level data for a DUX4 transcriptional target, ZSCAN4.

transcripts are similarly translated into stable proteins in FSHD muscle. However, further work is required to confirm this definitively. Similarly, our data strongly motivate focused investigation of these DUX4-induced proteins to test their utility as potential FSHD biomarkers in relevant clinical samples (*Yao et al., 2014*).

Next, we asked whether the changes in the DUX4 proteome are largely reflective of the changes to the transcriptome. We found that this is not the case. Although there is a positive correlation between these measurements (Pearson's correlation coefficient of 0.51), hundreds of genes deviate from this trend. GO analysis of the most differentially expressed genes at the transcript versus protein levels revealed RNA splicing and processing as the prominent categories impacted at the transcript level, whereas protein-level changes impacted an entirely different set of pathways. We take these results as an indication that the transcriptome-level analysis paints an incomplete picture of DUX4 biology, which should be complemented with proteome-level analysis to provide a thorough understanding of how DUX4 misexpression causes FSHD.

We next pursued the various mechanisms by which protein-level changes deviate from the corresponding transcriptomic changes. We found that genes that are induced by double-stranded RNA (dsRNA) and unfolded protein stress are transcriptionally induced, but translationally buffered as a result of the translational repression that accompanies activation of these stress-response pathways. Motivated by this result, we hypothesize that DUX4-expressing cells are unable to mount a robust stress response, despite inducing the transcripts necessary to alleviate stress. We also show that multiple proteins in RNA surveillance pathways, including UPF1 and XRN1, are down-regulated at the protein level, which may explain the drastic reduction in RNA quality control capacity that we observed in DUX4-expressing cells. From the proteomics data, we identified many genes that are involved in the ubiquitin proteasome pathway whose mis-regulation could alter protein stability. These genes may serve as a starting point for further investigation of how DUX4, which is best known as a transcriptional activator, induces widespread alterations in protein turnover.

# Materials and methods

## Key resources table

| Reagent type (species) or resource | Designation | Source or reference | Identifiers | Additional information |
|---|---|---|---|---|
| Cell line (*Homo sapiens*) | MB135 | PMID:28171552 | | |
| Cell line (*Homo sapiens*) | MB135-iDUX4 | PMID:28171552 | | |
| Antibody | Rabbit anti-GM130 antibody | Bethyl Laboratories Inc. | Cat # A303-402A-T | (1:250) |
| Antibody | Rabbit anti-Dux4 antibody | Abcam | Cat # ab124699 | (1:500) |
| Commercial assay, kit | ActinRed 555 ReadyProbes Reagent | ThermoFisher Scientific | Cat # R37112 | |
| Chemical compound, drug | DAPI | Sigma | Cat # D9542 | (1:1000) |
| Chemical compound, drug | L-LYSINE:2HCL UNLABELED | Cambridge Isotope Laboratories Inc. | Cat # ULM-8766-0.1 | |
| Chemical compound, drug | L-ARGININE:HCL UNLABELED | Cambridge Isotope Laboratories Inc. | Cat # ULM-8347-0.1 | |
| Chemical compound, drug | L-LYSINE:2HCL (13C6, 99%; 15N2, 99%) | Cambridge Isotope Laboratories Inc. | Cat # CNLM-291-H-0.05 | |
| Chemical compound, drug | L-ARGININE:HCL (13C6, 99%; 15N4, 99%) | Cambridge Isotope Laboratories Inc. | Cat # CNLM-539-H-0.05 | |
| Other | Ham's F12 for SILAC | Pierce | Cat # 88424 | |
| Other | Dialysed FBS for SILAC | Pierce | Cat # 88212 | |

## Cell culture and SILAC labeling

Proliferating human myoblasts (MB135) were cultured in F10 medium (Gibco/Life Technologies) supplemented with 20% fetal bovine serum (Thermo Scientific), 10ng bFGF (Life Technologies), 1µM dexamethasone (Sigma) and 50U/50µg penicillin/streptomycin (Life Technologies). We routinely confirmed the identity of these cells by PCR to amplify marker genes (DUX4 and/or Puromycin-resistance gene) as well as by tracking key morphological features that are characteristic of myoblast cells. These cells also tested negative for mycoplasma contamination. Cells were labeled in SILAC media containing heavy lysine (Lys8) and arginine (Arg10) for 3 weeks before the DUX4 induction experiments were carried out. To induce DUX4 expression in the MB135 iDUX4 cells, 1µg/ml of doxycycline was added for 8 or 14 hr, as indicated. For viral DUX4 expression, MB135 cells were transduced with lentivirus carrying DUX4 coding sequence under a hPGK promoter in the presence of polybrene.

## Gel slice digestion

Total RNA and protein were extracted from whole cells using TRIzol[RT] reagent (Ambion) following the manufacturer's instructions. 50 µg of total protein was subjected to SDS PAGE using a 4–15% bis-TRIS gel. The gel was stained using GelCode blue (Pierce) according to the manufacturer's instructions, destained overnight in ultrapure water and the entire lane containing the protein was cut into 16 fractions using a GelCutter (Gel company Inc.). Individual gel slices in 1.5 mL tubes (Eppendorf) were consecutively washed with water and incubated with 25mM ammonium bicarbonate in 50% acetonitrile for 2 hr. The gel pieces were dehydrated with acetonitrile, and the dried gel slices were reduced by covering them with 10 mM dithiothreitol in 100 mM ammonium bicarbonate and heating them at 56°C for 45 min. The solution was removed and discarded. The gel slices were alkylated by covering them with a solution of 50 mM iodoacetamide in 100 mM ammonium bicarbonate and incubating in the dark at ambient temperature for 30 min. The solution was removed and discarded. The gel slices were dehydrated with acetonitrile, then washed with 100 mM ammonium bicarbonate for 10 min. The solution was removed, discarded and the gel slices were

dehydrated once again with acetonitrile. After removing acetonitrile, the gel slices were then hydrated with 5 ng/uL sequencing-grade trypsin (Promega) in 50 mM ammonium bicarbonate and digested overnight at 37°C on an orbital shaker. Following digestion, the supernatants were collected, and the gel slices were washed with 0.1% trifluoroacetic acid, and after 30 min an equal volume of acetonitrile was added followed by washing for an additional 1 hour. The original digestion supernatant and the wash for a single sample were combined into a single tube and dried by vacuum centrifugation. The digestion products were desalted using Ziptips (Millipore) according to the manufacturer's instructions, eluted with 70% acetonitrile/0.1% trifluoroacetic acid, and dried by vacuum centrifugation.

## Mass spectrometry

The desalted material was resuspended in 20 μL of 2% acetonitrile in 0.1% formic acid, and 18 μL was analyzed using one of two LC/ESI MS/MS configurations. The first configuration consisted of an Easy-nLC II (Thermo Scientific) coupled to a Orbitrap Elite ETD (Thermo Scientific) mass spectrometer using a trap-column configuration as described (*Licklider et al., 2002*). A trap of 100 μm × 20 mm packed with Magic $C_{18}$AQ (5-μm, 200 Å resin; Michrom Bioresources) packing material was used for in-line desalting and a column of 75 μm × 250 mm packed with $C_{18}$AQ (5-μm, 100 Å resin; Michrom Bioresources) was used for analytical peptide separations. Chromatographic separations were carried out using a 60-minute gradient from 5% to 35% solvent B (solvent A: 0.1% formic acid, solvent B: 0.1% formic acid in acetonitrile) at a flowrate of 300 nL/min. The analytical column temperature was maintained at 40°C. The Orbitrap Elite instrument was operated in the data-dependent mode, switching automatically between MS survey scans in the Orbitrap (AGC target value 1E6, resolution 240,000, and maximum injection time 250 ms) and collision induced dissociation (CID) MS/MS spectra acquisition in the linear ion trap (AGC target value of 10,000 and injection time 100 ms). The 20 most intense precursor ions from the OrbiTrap full scan were each consecutively selected for fragmentation by CID in the linear ion trap using a normalized collision energy of 35%. Ions of +2 and +3 charge states were selected for MS/MS and selected ions were dynamically excluded for 30 seconds. The second configuration consisted of an Easy nanoLC 1000 (Thermo Scientific) HPLC connected to an OrbiTrap Fusion (Thermo Scientific) mass spectrometer. In-line chromatographic separations (no trap column) were carried out using a 75 μm × 400 mm column packed with Magic $C_{18}$AQ (5-μm, 100 Å resin; Michrom Bioresources) packing material at a flowrate of 300 nL/min. Chromatographic elution consisted of a 90-minute gradient from 3% to 27% solution B and the column temperature was maintained at 40°C. The OrbiTrap Fusion was operated in the 2 sec 'top speed' data dependent acquisition mode with MS survey scans in the OrbiTrap at least every 2 sec (AGC target value 4E5, resolution 120,000, and maximum injection time of 50 ms). Quadrupole isolation was set to 1.6 full width at half maximum (FWHM) and higher energy collision dissociation (HCD) was used for fragmentation at a collision energy of 28%. MS/MS detection was carried out in the linear ion trap set at rapid scan speed (injection time of 250 ms and AGC target of 10E2). Positively charged ions from 2 to 6 were selected for MS/MS and selected ions were dynamically excluded for 30 sec.

## Data analysis and statistical methods

Qualitative and quantitative data analysis were performed using Proteome Discoverer 2.1 (Thermo Scientific). The data were searched against a human UniProt database (downloaded 11-04-16) that was appended with protein sequences from the common Repository of Adventitious Proteins (cRAP; www.thegpm.org/crap/) and *in silico* translation products of noncanonical transcript isoforms stabilized as the result of NMD inhibition. In downstream analyses, peptides that only mapped to NMD targets were not considered any further in the current study and will be pursued in a future investigation. Searches were conducted with the trypsin enzyme specificity. The precursor ion tolerance was set to 10 ppm and the fragment ion tolerance was set to 0.6 Da. Variable modifications were set for oxidation on methionine (+15.995 Da), carbamidomethyl (+57.021 Da) on cysteine, and acetylation (+42.010 Da) on the N-terminus of proteins. Heavy SILAC amino acids for lysine (+8.014 Da) and arginine (10.008) were also accounted for in the analysis as variable modifications. All search results were evaluated by Percolator (*Käll et al., 2007*) for false discovery rate (FDR) evaluation of the identified peptides. Peptide identifications were filtered to a peptide FDR of 1%.

The peptide-spectrum matches (PSMs) and the corresponding quantification data were then exported as a tab-delimited text file from Proteome Discoverer 2.1, and all downstream data analysis was conducted using the R statistical programming language. The Custom R script (Source Code File 1) to reproduce the analyses described above and figures in this manuscript is deposited in github (*Jagannathan et al., 2019*; copy archived at https://github.com/elifesciences-publications/2019-eLife-jagannathan_et_al). Briefly, we filtered PSMs to remove the spectra carrying the following quality flags assigned by Proteome Discoverer – 'Inconsistently labeled', 'Indistinguishable channels', 'No quantitative values', 'Not unique', 'Redundant spectra', 'Single peak spectra', and 'Excluded by method' – all of which represent poor quality spectra. Next, we removed peptides that failed to map to a gene, and peptides mapping to genes in the cRAP database or to uncharacterized genes. We then assigned the median peptide fold change within an experiment as the summarized protein fold change value.

Upon comparing the gene-level fold changes obtained from four different samples, we noticed a strong correlation between the two viral replicates (Pearson's $r = 0.81$), as well as the data from iDUX4 replicate 1 and both vDUX4 datasets (Pearson's $r = 0.79$ and 0.77). The iDUX4 replicate 2, in which we swapped the isotopic labels used for the control and DUX4 samples, however, showed poor correlation with the three other samples because of a small set of proteins (103 proteins; 2.6% of total) that did not label efficiently. Proteins that do not label efficiently result in apparent anticorrelation between label-swap samples. This issue, wherein a small subset of proteins does not label efficiently, is a well-known problem in SILAC experiments (*Park et al., 2012*). Rather than ignoring this subset of proteins, which would require the imposition of potentially arbitrary thresholds, we instead followed the standard SILAC analysis procedure of taking the median value of H/L ratio across the two iDUX4 replicates as the final iDUX4 H/L ratio (*Park et al., 2012*; *Ong and Mann, 2006*). This procedure preserves the H/L ratio for proteins that labeled efficiently (the vast majority of the proteome) while bringing the H/L ratio for proteins that did not label efficiently to approximately 1 (corresponding to no measurable change). We then calculated *p*-values for the fold change calculated for the combined iDUX4 samples using a bootstrap approach. For a protein with n peptides whose H/L ratios were used to calculate protein fold change, we calculated the probability that n peptides chosen randomly out of all measured peptides would yield a similar fold change by chance. To this end, we generated a null distribution for each protein as follows. First, the same number of H/L ratios as the number of peptides quantified for that protein were randomly chosen from the set of all calculated peptide-level H/L ratios (n=229,558). With these randomly chosen H/L ratios, a fold change was calculated by computing the median. This process was repeated 1,000 times to generate the null distribution of H/L ratios for that protein. Using this null distribution, a *p*-value was calculated using a two-tailed Z test.

## Gene Ontology analysis

GO analysis was performed via the Overrepresentation Enrichment Analysis method using WebGestalt server (pmid:15980575; www.webgestalt.org). One of the GO categories identified as enriched in this analysis was skin development, which we subsequently removed from *Figure 2C* as many of the genes that contribute to this GO category were extracellular proteins including keratins that could be environmental contaminants.

## Immunofluorescence microscopy

Cells were permeabilized with PBS containing 0.1% Triton X-100 for 5 min at room temperature and rinsed three times in PBS. Primary antibody against GM130 (Bethyl Laboratories, Cat # A303-402A-T) was diluted 1:200 in PBS and incubated for 1 hr at room temperature. After three washes in PBS, secondary anti-Rabbit TRITC (Jackson ImmunoResearch, cat # 711-025-152) diluted 1:400 was added and incubated for 45 min at room temperature. Cells were washed three times in PBS with the nuclear counterstain DAPI included in the final wash. Images were collected on a Cytation 5 multi-mode reader (BioTek) and analyzed using GenPrime software (BioTek).

## Acknowledgements

The authors wish to acknowledge Heidi Dvinge for helpful discussions on proteomics data analysis. We also thank current and past members of the Bradley and Tapscott laboratories for comments

and suggestions that have improved this body of work over time. This research was supported by the NINDS P01NS069539 (RKB and SJT) and the FSH Society FSHS-22014-01 (SJ). RKB is a Scholar of The Leukemia & Lymphoma Society. The Fred Hutchinson Cancer Research Center Proteomics Facility is supported by a National Institutes of Health Cancer Center Support Grant (P30 CA015704). The OrbiTrap Fusion mass spectrometer used in this research was purchased with a grant from the MJ Murdock Charitable Trust.

## Additional information

### Funding

| Funder | Grant reference number | Author |
| --- | --- | --- |
| National Institute of Neurological Disorders and Stroke | P01NS069539 | Stephen J Tapscott Robert K Bradley |
| FSH Society | FSHS-22014-01 | Sujatha Jagannathan |
| Leukemia and Lymphoma Society | | Robert K Bradley |
| National Institutes of Health | P30 CA015704 | Yuko Ogata Philip R Gafken |
| M.J. Murdock Charitable Trust | | Yuko Ogata Philip R Gafken |

The funders had no role in study design, data collection and interpretation, or the decision to submit the work for publication.

### Author contributions

Sujatha Jagannathan, Conceptualization, Resources, Data curation, Software, Formal analysis, Funding acquisition, Validation, Investigation, Visualization, Methodology, Writing—original draft, Writing—review and editing; Yuko Ogata, Data curation, Writing—review and editing, Mass spectrometry data acquisition; Philip R Gafken, Data curation, Supervision, Funding acquisition, Methodology, Writing—review and editing, Mass spectrometry data acquisition; Stephen J Tapscott, Robert K Bradley, Conceptualization, Resources, Supervision, Funding acquisition, Project administration, Writing—review and editing

### Author ORCIDs

Sujatha Jagannathan http://orcid.org/0000-0001-9039-2631
Stephen J Tapscott https://orcid.org/0000-0002-0319-0968
Robert K Bradley https://orcid.org/0000-0002-8046-1063

### Decision letter and Author response

Decision letter https://doi.org/10.7554/eLife.41740.sa1
Author response https://doi.org/10.7554/eLife.41740.sa2

## Additional files

### Supplementary files

• Supplementary file 1. Gene-level log2 fold change values for RNA and protein in iDUX4 MB135 cells.

• Transparent reporting form

### Data availability

Mass spectrometry proteomics data have been deposited to the ProteomeXchange Consortium via the PRIDE partner repository (Vizcaino et al., 2013) with the dataset identifier PXD010221. To enable easy access to processed peptide-spectrum match data, we have also deposited peptide-level data

to Dryad (doi:10.5061/dryad.ck06k75). The previously published RNA-seq data are available through the NCBI SRA database under accession number GSE85461.

The following datasets were generated:

| Author(s) | Year | Dataset title | Dataset URL | Database and Identifier |
|---|---|---|---|---|
| Jagannathan S, Ogata Y, Gafken PR, Tapscott SJ, Bradley RK | 2018 | Proteomics analysis of DUX4-expressing myoblasts | https://www.ebi.ac.uk/pride/archive/projects/PXD010221 | PRIDE PRoteomics IDEntifications database, PXD010221 |
| Jagannathan S, Ogata Y, Gafken P, Tapscott S, Bradley R | 2018 | Data from: Quantitative proteomics reveals key roles for post-transcriptional gene regulation in the molecular pathology of FSHD. | https://dx.doi.org/10.5061/dryad.ck06k75 | Dryad Digital Repository, 10.5061/dryad.ck06k75 |

The following previously published dataset was used:

| Author(s) | Year | Dataset title | Dataset URL | Database and Identifier |
|---|---|---|---|---|
| Jagannathan S | 2016 | Model systems of DUX4 expression recapitulate the transcriptional profile of FSHD cells | https://www.ncbi.nlm.nih.gov/geo/query/acc.cgi?acc=GSE85461 | NCBI Gene Expression Omnibus, GSE85461 |

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
