## [Decision Letter]

Thank you for submitting your article "Quantitative proteomics reveals key roles for post-transcriptional gene regulation in the molecular pathology of FSHD" for consideration by *eLife*. Your article has been reviewed by two peer reviewers, and the evaluation has been overseen by a Reviewing Editor and Detlef Weigel as the Senior Editor. The following individual involved in review of your submission has agreed to reveal his identity: Andrew Berglund (Reviewer #3).

The reviewers have discussed the reviews with one another and the Reviewing Editor has drafted this decision to help you prepare a revised submission.

Summary:

The study reveals the many proteomic changes caused by expression of DUX4. The comparison with transcriptomic data is important because it demonstrates that while there is a correlation between transcript and protein level changes (about 0.5), some protein levels changes are not predicted by changes in gene expression. In particular, it appears that post-transcriptional buffering of the stress-response pathway may not allow this pathway to operate in DUX4 induced cells. Post-transcriptional regulation also is important in the NMD pathway – building on previous findings from the Bradley and Tapscott groups. A nice web tool was built for others to access the data.

Essential revisions:

1) An interesting observation is the down-regulation of several factors at the protein level that is required for mRNA surveillance and turnover, including UPF1, UPF3B, and XRN1. The authors previously reported in an *eLife* paper published in 2015 that expression of aberrant RNAs, including NMD substrates, is a characteristic of DUX4-expressing cells. It would therefore add considerably to their present study if they could use their SILAC MS data to assess the extent to which these transcriptomic changes correspond to the expression of aberrant protein products, and also describe possible examples in which this may provide novel insight into FSHD pathology.

2) It is apparent that the authors used a myoblast cell line for their experiments, but details are not provided. Is it the same MB135 line as used in their previous study? To what extent do the authors think they can extrapolate their findings to the pathophysiology of FSHD in skeletal muscle tissue? Are there available patient and/or FSHD mouse model samples with which they might begin to address this question? The limitations of using a cell line in their study should at least be discussed.

3) The authors performed their iDUX and vDUX4 experiments at different time points and using different instruments yet state that one served as a biological replicate for the other. It is unclear to this reviewer the overall degree of correlation between the changes observed using the two experimental approaches, nor how these compare to replicates performed for each protocol. It is also unclear in some cases whether protein level changes highlighted by the authors are consistently observed between the two approaches. Please clarify these details.

4) Related to the last comment, some aspects of the methods are vague or difficult to follow, particularly relating to MS data quantification and analyses. The authors should provide enough detail for the reader to follow and also explain their choice of methods for analyzing SILAC data. Also, many of the legends and the associated main text do not provide sufficient information to follow how various plots were generated. Figures 1B-E are examples. The authors should take particular care to expand their legends and descriptions of analyses to address this concern.

5) In terms of usefulness for potential biomarkers, it would be important to demonstrate that these findings in a cell model translate to clinical samples. To significantly strengthen the results from this work it would be great to see some of these findings recapitulated in samples from patients. It might be a very challenging request due to lack of samples or that very few cells express DUX4 in the disease state. If no relevant clinical samples can be obtained, the authors should be careful with how they describe the usefulness of the results in terms of biomarkers.

---

## [Author Response]

Essential revisions:1) An interesting observation is the down-regulation of several factors at the protein level that is required for mRNA surveillance and turnover, including UPF1, UPF3B, and XRN1. The authors previously reported in an eLife paper published in 2015 that expression of aberrant RNAs, including NMD substrates, is a characteristic of DUX4-expressing cells. It would therefore add considerably to their present study if they could use their SILAC MS data to assess the extent to which these transcriptomic changes correspond to the expression of aberrant protein products, and also describe possible examples in which this may provide novel insight into FSHD pathology.

We agree that it is interesting and important to determine whether DUX4-induced suppression of mRNA quality control results in the production of aberrant proteins. Because mRNA translation is maintained after DUX4 expression (Figure 3C-D), it is reasonable to hypothesize that DUX4-expressing cells produce aberrant proteins that may contribute to the inflammatory phenotype of FSHD. However, as described below, obtaining direct evidence of such aberrant proteins from our SILAC data is challenging due to the limited coverage of proteomics data.

To address this comment, we conducted a focused analysis of our SILAC data to search for peptides arising from mRNAs that are normally degraded by nonsense-mediated decay (NMD). We successfully detected a number of peptides encoded by NMD substrates. However, we were unable to accurately measure peptide abundance or detect the same peptide across replicates. We believe that this is due to the low abundance of such “NMD peptides” as well as the inherent challenge of using proteomics to identify aberrant protein isoforms whose sequences are highly similar to much more abundant “normal” isoforms (e.g., an “NMD peptide” typically differs from its non-NMD counterpart by only 10-20aa at its C terminus).

We conducted a theoretical analysis (schematic in Author response image 1) of all peptides that could arise from trypsin digestion of the theoretical human proteome (including NMD-targeted aberrant transcripts). This analysis revealed that only 3,568 peptides out of a total of 3,282,166 peptides (0.12%) are uniquely encoded by NMD substrate mRNAs. Of all possible peptides from the human proteome, we observed a total of ~45,000 unique peptides (1.3% of total peptides) after combining data from all of our SILAC samples (iDUX4, vDUX4, and the corresponding controls). This level of statistical power implies that we can only expect to observe a total of ~50 NMD peptides after combining data across all of our experiments. This theoretical calculation is consistent with our sporadic observation of NMD peptides in our data, but at such a low level that we cannot measure their abundance

Given these inherent limitations of proteomics data, we believe that a different technology is most appropriate for detecting translation of aberrantly stabilized NMD substrate mRNAs. We are currently testing the suitability of ribosome profiling for this assay, but respectfully feel that these experiments are beyond the scope of the current manuscript.

2) It is apparent that the authors used a myoblast cell line for their experiments, but details are not provided. Is it the same MB135 line as used in their previous study?

Yes, we used the same MB135 cell line that was used in the previous study (Feng et al., 2015). We apologize for this omission. We provide this information in the revised manuscript, both in the Results and the Materials and methods sections, as follows.

Results:

“In order to measure DUX4-induced changes to the cellular proteome, we conducted SILAC-based mass spectrometry in two independent DUX4 expression systems (Figure 1A). […] Here, we used both of these expression systems in order to internally corroborate our results, thereby ensuring that our proteomic data were robust with respect to choice of model system.”

Materials and methods:

“To induce DUX4 expression in the MB135 iDUX4 cells, 1µg/ml of doxycycline was added for 8 or 14 hours, as indicated. For viral DUX4 expression, MB135 cells were transduced with lentivirus carrying DUX4 coding sequence under a hPGK promoter in the presence of polybrene.”

To what extent do the authors think they can extrapolate their findings to the pathophysiology of FSHD in skeletal muscle tissue? Are there available patient and/or FSHD mouse model samples with which they might begin to address this question? The limitations of using a cell line in their study should at least be discussed.

In addition to revealing the DUX4-induced proteome, a central goal of our study was to validate the utility of FSHD molecular biomarkers that we previously proposed (Figure 1D-E in this study; Yao et al., 2014). We originally identified these biomarkers by synthesizing transcriptome analyses of cell culture models of DUX4 expression with primary patient materials. These biomarkers are robustly up-regulated in FSHD relative to healthy control muscle and their expression correlates with clinical severity (Yao et al., 2014). Therefore, our report in the current study that these biomarkers are robustly up-regulated at both the mRNA and protein levels strongly suggests that our cell culture-based proteomics reveals important molecular features of FSHD.

The current study was necessarily limited to cell culture due to our use of the SILAC (Ong et al., 2002) technology to obtain quantitative estimates of protein abundance. While cell culture-based models of DUX4 expression cannot capture all aspects of FSHD pathophysiology, we previously demonstrated that our vDUX4 and iDUX4 cell culture systems accurately recapitulated the transcriptional profile of FSHD (Jagannathan et al., 2016). We therefore believe that our proteomics data likely captures molecular dysregulation that occurs in primary patient samples, although further work is required to confirm this.

We agree that it would be interesting to extend our studies to a murine system. However, despite substantial effort from several groups, including ours, there is no single, widely accepted mouse model of FSHD. The challenge of developing a mouse model that accurately captures key disease features may arise from the absence of *DUX4* from the mouse genome. The mouse *Dux* gene arose from an ancestral gene duplication and has a different DNA-binding profile and transcriptional program than does human *DUX4* (Whiddon et al., 2017).

The revised manuscript includes the following statement in the Discussion to address the reviewer’s point above:

“Here, we used quantitative mass spectrometry in two different cell culture models of FSHD to demonstrate that DUX4-induced transcripts are efficiently translated into stable proteins in muscle cells. […] Similarly, our data strongly motivate focused investigation of these DUX4-induced proteins to test their utility as potential FSHD biomarkers in relevant clinical samples.”

3) The authors performed their iDUX and vDUX4 experiments at different time points and using different instruments yet state that one served as a biological replicate for the other.

We apologize for our confusing terminology. We used the word “replicate” to refer to these experiments, which are actually quite different (as the reviewer points out), in order to emphasize the high reproducibility that we observed between distinct experiments. In the revised manuscript, we clearly state that our data arose from independent experiments and samples that corroborate each other, rather than describing these data as arising from biological replicates.

Results:

“We previously showed that DUX4 expressed via a lentiviral vector versus an inducible transgene integrated into the genome of a myoblast cell line both yield comparable gene expression profiles that accurately capture the transcriptome of FSHD cells (Jagannathan et al., 2016). Here, we used both of these expression systems in order to internally corroborate our results, thereby ensuring that our proteomic data were robust with respect to choice of model system.”

It is unclear to this reviewer the overall degree of correlation between the changes observed using the two experimental approaches, nor how these compare to replicates performed for each protocol.

We observed excellent correlations between our two different experimental approaches and replicates, with one caveat that is inherent to SILAC experiments (described in more detail below). It is straightforward to assess the correlation between the two viral replicates (Pearson’s *r* = 0.81), as well as the data from iDUX4 replicate 1 and both vDUX4 datasets (Pearson’s *r* = 0.79 and 0.77; see Author response image 2 for an example).

**Author response image 2. respfig2:** 

The analysis is more complex for the comparison between the first and second iDUX4 replicates. These replicates correspond to a label-swap experiment, in which we swapped the labels used for the control and DUX4 samples. For this experiment, there was a small set of genes (103 genes; 2.6% of total) that did not label efficiently, and so exhibited anticorrelation between the samples. This issue, wherein a small subset of proteins does not label efficiently, is a well-known problem in SILAC experiments (e.g., Park et al., 2012). Rather than simply ignore this subset of genes, which would require the imposition of potentially arbitrary thresholds, we instead followed the standard SILAC analysis procedure of taking the median value of H/L ratio across the two iDUX4 replicates as the final iDUX4 H/L ratio (Ong and Mann, 2006). This procedure preserves the H/L ratio for proteins that labeled efficiently (the vast majority of the proteome) while bringing the H/L ratio for proteins that did not label efficiently to approximately 1 (corresponding to no measurable change). In the revised manuscript, we discuss this phenomenon and our method for handling it in the expanded:

Materials and methods section:

“Upon comparing the gene-level fold changes obtained from four different samples, we noticed a strong correlation between the two viral replicates (Pearson’s *r* = 0.81), as well as the data from iDUX4 replicate 1 and both vDUX4 datasets (Pearson’s *r* = 0.79 and 0.77). […] This procedure preserves the H/L ratio for proteins that labeled efficiently (the vast majority of the proteome) while bringing the H/L ratio for proteins that did not label efficiently to approximately 1 (corresponding to no measurable change).”

It is also unclear in some cases whether protein level changes highlighted by the authors are consistently observed between the two approaches. Please clarify these details.

We apologize for the confusion. The protein level changes that we highlighted were consistent between the two approaches (vDUX4 and iDUX4), with the caveat that some proteins were only quantified in the iDUX4 dataset due to its higher coverage and depth. Our web server permits readers to readily view raw data for any protein of interest, which will allow them to examine the consistency and robustness of specific fold-change estimates.

4) Related to the last comment, some aspects of the methods are vague or difficult to follow, particularly relating to MS data quantification and analyses. The authors should provide enough detail for the reader to follow and also explain their choice of methods for analyzing SILAC data.

We apologize for not providing a sufficiently full description of our methods for MS data quantification and analysis. In the revised manuscript, we addressed this concern in three ways.

First, our methods are described as comments in the open-source software that we wrote to analyze our data (Supplementary File 3).

Second, we added the following text to the Materials and methods section:

“The peptide-spectrum matches (PSMs) and the corresponding quantification data were then exported as a tab-delimited text file from Proteome Discoverer 2.1, and all downstream data analysis was conducted using the R statistical programming language. […] This process was repeated 1,000 times to generate the null distribution of H/L ratios for that protein. Using this null distribution, a *p*-value was calculated using a two-tailed Z test.”

Finally, we have added additional information regarding data analysis to Figure 1A, our schematic representation of the experimental strategy.

Also, many of the legends and the associated main text do not provide sufficient information to follow how various plots were generated. Figures 1B-E are examples. The authors should take particular care to expand their legends and descriptions of analyses to address this concern.

We apologize for this lack of clarity. In the revised manuscript, we addressed this concern by expanding the descriptions of our experiments as well as the corresponding legends.

5) In terms of usefulness for potential biomarkers, it would be important to demonstrate that these findings in a cell model translate to clinical samples. To significantly strengthen the results from this work it would be great to see some of these findings recapitulated in samples from patients. It might be a very challenging request due to lack of samples or that very few cells express DUX4 in the disease state. If no relevant clinical samples can be obtained, the authors should be careful with how they describe the usefulness of the results in terms of biomarkers.

We agree that it would be ideal to confirm our cell culture-based measurements of protein abundance in primary patient materials. However, doing so is very challenging due to unusual features of DUX4 and FSHD biology. DUX4 is only expressed in ~0.1% of FSHD cells (Snider et al., 2010), rendering direct detection of a DUX4-induced protein extremely difficult. For perspective, it is not yet possible to detect DUX4 protein itself via IHC in affected patient muscle. Because of DUX4’s sporadic and variegated expression pattern in FSHD cells, most of our understanding of DUX4-induced molecular dysregulation relies upon cell culture models. We and others have therefore devoted substantial effort to developing cell culture systems that accurately capture key molecular features of FSHD cells, such as our vDUX4 and iDUX4 systems (Jagannathan et al., 2016). We therefore believe that our proteomics data likely captures molecular dysregulation that occurs in primary patient samples, although we agree with the reviewer that further work is required to confirm this.

As the reviewer points out, an important goal of our study was to validate the utility of FSHD molecular biomarkers that we previously proposed (Figure 1D-E in this study; Yao et al., 2014). We originally identified these biomarkers by synthesizing transcriptome analyses of cell culture models of DUX4 expression with primary patient materials. These biomarkers are robustly up-regulated in FSHD relative to healthy control muscle and their expression correlates with clinical severity (Yao et al., 2014). However, prioritizing specific biomarkers for protein assay development in primary patient materials has been challenging because protein production had not been systematically tested. We believe that our current study provides this critical missing link.

The revised manuscript includes the following statement in the Discussion to address the reviewer’s point above:

“Here, we used quantitative mass spectrometry in two different cell culture models of FSHD to demonstrate that DUX4-induced transcripts are efficiently translated into stable proteins in muscle cells. […] Similarly, our data strongly motivate focused investigation of these DUX4-induced proteins to test their utility as potential FSHD biomarkers in relevant clinical samples (Yao et al., 2014).”

References:

Ong SE, Blagoev B, Kratchmarova I, Kristensen DB, Steen H, Pandey A, Mann M. (2002) Stable isotope labeling by amino acids in cell culture, SILAC, as a simple and accurate approach to expression proteomics. Mol Cell Proteomics. 1(5):376-86.

Snider L, Geng LN, Lemmers RJ, Kyba M, Ware CB, Nelson AM, Tawil R, Filippova GN, van der Maarel SM, Tapscott SJ, Miller DG. (2010) scapulohumeral dystrophy: incomplete suppression of a retrotransposed gene.

PLoS Genet. 6(10):e1001181. doi: 10.1371/journal.pgen.1001181.

Whiddon JL, Langford AT, Wong CJ, Zhong JW, Tapscott SJ. (2017) Conservation and innovation in the DUX4-family gene network. Nat Genet. 49(6):935-940. doi: 10.1038/ng.3846.